# Synthesis and Properties of NitroHPHAC: The First Example of Substitution Reaction on HPHAC

**DOI:** 10.3390/molecules25112486

**Published:** 2020-05-27

**Authors:** Yoshiki Sasaki, Masayoshi Takase, Shigeki Mori, Hidemitsu Uno

**Affiliations:** Graduate School of Science and Engineering, Ehime University, Matsuyama 790-8577, Japan; f865001a@mails.cc.ehime-u.ac.jp.com (Y.S.); mori.shigeki.mu@ehime-u.ac.jp (S.M.)

**Keywords:** hexapyrrolohexaazacoronene, nitration, S_N_Ar substitution, ICT character, aromaticity

## Abstract

Hexapyrrolohexaazacoronene (HPHAC) is one of the N-containing polycyclic aromatic hydrocarbons in which six pyrroles are fused circularly around a benzene. Despite the recent development of HPHAC analogues, there is no report on direct introduction of functional groups into the HPHAC skeleton. This work reports the first example of nitration reaction of decaethylHPHAC. The structures of nitrodecaethylHPHAC including neutral and two oxidized species (radical cation and dication), intramolecular charge transfer (ICT) character, and global aromaticity of the dication are discussed.

## 1. Introduction

Introduction of functional groups into π-conjugated systems is a straightforward way to tune their chemical and physical properties. A nitro group is one of the most useful groups for such purpose. Aromatic nitro compounds have been studied and utilized for medicines, pesticides, dyes, pigments, plastics raw materials, and so on. The functional group strongly stabilizes electron-rich polycyclic aromatic hydrocarbons (PAHs) and can be transformed to many other functional groups by simple reactions [1]. When the nitro group is introduced to large π-conjugated compounds, the LUMO levels would be sufficiently lowered so that nucleophiles could directly attack the π-conjugated systems. The nitro group can be easily transformed into an amino group, which is recognized as one of the most powerful electron-donating groups, as well. Therefore, nitration reaction is one of the first choices for further derivatization of aromatic compounds. Recently, large π-conjugated materials with a nitro group and their derivatives have attracted continuing attention from many research areas [2,3,4,5,6,7,8,9,10,11,12,13,14].

Hexapyrrolohexaazacoronene (HPHAC) is a nitrogen-embedded PAH, consisting of circularly connected pyrroles around a benzene core and is easily prepared from hexapyrrolylbenzene by the Scholl oxidation using FeCl_3_ [15], 2,3-dichloro-5,6-dicyano-1,4-benzoquinone (DDQ) in the presence of trifluoromethanesulfonic acid [16], or *N*-bromosuccinimide (NBS) [17]. The HPHAC π-system consists of local π-systems of six pyrroles and one benzene and has a non-aromatic character in the whole molecule. Due to the pyrrole moieties, on the other hand, multiple oxidation levels of HPHAC are reversibly generated, and its dication shows global aromaticity owing to the cyclic conjugation of a 22π-electron system. So far, HPHAC-hexabenzocoronene-hybridized [18], ethylene-bridged [19], periphery-expanded [20,21], core-expanded [16], azulene-fused [22], β,β-thieno-fused [23], chiral [24], antiaromatic [25], and σ-dimerized [26] types of its analogues have been reported. However, a peripheral substitution reaction of HPHAC has never been reported yet. Functionalization of HPHACs has all been dependent on the composed pyrroles employed. Herein, we report the first example of nitration on decaethylHPHAC (DEHPHAC **1a**), leading to nitroDEHPHAC **2a** (Scheme 1). The fundamental properties of nitroDEHPHAC **2a** were investigated by NMR and absorption spectroscopy, single crystal X-ray structure analysis, and theoretical calculations.

## 2. Results and Discussion

DEHPHAC **1a** was synthesized from *N*-(2,3,4,5,6-pentafluorophenyl)-1*H*-pyrrole via decaethylated hexapyrrolylbenzene DEHPB by S_N_Ar and successive Scholl reactions in accordance with the previously reported procedure [15,27]. The nitration of **1a** was then carried out with silver nitrite. In contrast to the cases of corrole and porphyrin [28,29], DEHPHAC **1a** did not dimerize by this oxidant but was transformed into nitroDEHPHAC **2a** in a 51% yield (Scheme 1). The nitration mechanism is not unclear at this moment although we suppose a coupling of cation radical **2a**^•+^ with nitrogen dioxide (•NO_2_) plays a key role. The crude product was purified by silica-gel column chromatography with hexane and toluene as the eluent. The structure of **2a** was unambiguously identified by ^1^H-NMR, high-resolution LDI-TOF MS, and X-ray single-crystal structure analyses.

The ^1^H-NMR spectrum of **2a** exhibited signals due to one β-proton of pyrrole and fifty protons of ethyl groups, and these signals appeared in the lower fields compared to those of **1a** due to an electron-withdrawing effect of the nitro group. Single crystals were fortunately obtained by vapor diffusion of methanol into the toluene solution of **2a**. The X-ray crystal structure analysis revealed the averaged bond length of the N-O moiety is 1.233(6) Å, the value of which is similar to that of nitrobenzene (1.226 Å) [30]. The molecules in the crystal adopted a slipped columnar structure with π–π stacking distances of 3.406 and 3.449 Å, calculated by the mean planes of 24 atoms forming hexaazacoronene. The values are almost the same as the sum of van der Waals (vdW) radii of carbon atoms (ca. 3.4 Å) (Figure 1A,B).

To investigate the intramolecular charge transfer (ICT) character between the electron-rich HPHAC moiety and the nitro group, the absorption spectra of **1a** and **2a** were considered by the comparison with density functional theory (DFT) and the time-dependent density functional theory (TD-DFT) calculations of decamethylHPHACs (DMHPHACs **1b** and **2b**), where all ethyl groups of **1a** and **2a** were replaced by methyl groups for simplicity (Appendix A). The cut-off of **2a** in a CH_2_Cl_2_ solution is bathochromically shifted by ca. 200 nm compared to that of **1a** (Figure 2). A characteristic broad band around 540–740 nm is ascribed to the mixed transitions of HOMO–LUMO and HOMO−1–LUMO, according to TD-DFT calculations (Appendix A). Figure 3 shows the molecular orbitals (MOs) of **1b** and **2b**. By the introduction of a nitro group on DMHPHAC **1b**, both HOMO and LUMO levels are lowered, and the decrease of the LUMO energy is larger than that of HOMO (*△E* = 0.86 for LUMO and 0.39 eV for HOMO). The LUMO is derived from the nitro group, mainly located at the nitro group, and slightly developed to the DMHPHAC skeleton. The LUMO+1 of **2b** is correlated to the LUMO of **1b,** and the LUMO+2 and LUMO+3 of **2b** are from degenerated LUMO+1 and LUMO+2 of **1b**. As a result, the HOMO–LUMO gap of nitroDMHPHAC **2b** is narrower than that of **1b**, which is attributable to the broad absorption band in the longest wavelength region of the ICT transition. In fact, positive solvatochromism was observed for **2a** in various solvents (Appendix A) because of strong stabilization of polarized LUMO of **2a** by polar solvents. NitroDEHPHAC **2a** showed little emission.

Electrochemical properties of **2a** were examined by cyclic and differential pulse voltammograms (CV and DPV) (Figure 4). CV of **2a** showed two reversible oxidation waves at −0.20 and 0.07 V against a ferrocene/ferrocenium ion couple, values of which are low compared to those of **1a**. Thus, oxidative titration of **2a** was performed with pentachloroantimony(V) as monitoring the absorption spectra and resulted in the two-step changes with isosbestic points from neutral to radical cationic and then to dicationic species (Figure 5 and Appendix A). The peak maxima of radical cationic and dicationic species appeared in the visible to NIR region at 627 and 688 nm together with a broad band centered at ca. 1150 nm and at 784 and 884 nm with a broad band centered at ca. 1450 nm, respectively, which are characteristic for the previously reported the HPHAC radical cations and dications [15].

For better understanding of the structures and electronic properties of oxidized species, the radical cation and dication of **2a** were aimed to be prepared by controlling the amount of an oxidant employed (Scheme 1). Both species were quantitatively obtained by the oxidation of proper amounts of silver(I) hexafluorophosphate as the oxidant, successfully isolated as hexafluorophosphate salts (isolated yields were not determined) and stable enough to handle under air in the solid states. Single crystals of **2a**^•+^[PF_6_^−^] were obtained by slow vapor diffusion of hexane into the chlorobenzene solution and were subject to X-ray analysis. The crystal turned out to be composed of one molecule of **2a**^•+^[PF_6_^−^] and one and a half molecules of chlorobenzene in the asymmetric unit, and the radical cation molecule of **2a**^•+^ was revealed to form a π-dimeric structure by an inversion center in the unit cell (Figure 6 and Appendix A). The closest atoms between the neighboring molecules in the π-dimer faced by the convex side are found in the benzene core, and the distance is 3.173 Å. The distance of the mean planes of 24 hexaazacoronene atoms was 3.348 Å. These values are shorter than the sum of vdW radii of carbon atoms, strongly suggesting the π-dimer formation between two radical cations [31,32,33,34].

The X-ray crystal structure of dication **2a**^2+^[PF_6_^−^]_2_ was also determined and illustrated in Appendix A. In general, the bond-length alternation (BLA) is important to discuss the nature of π-system. Table 1 shows the selected bond lengths of the pyrrole rings of neutral **2a**, radical cation **2a**^•+^, and dication **2a**^2^⁺. By increasing the number of oxidation states, the averaged bonds of C*_α_*-C*_α_* and C*_β_*-C*_β_* become shorter, and the bond of C*_α_*-C*_β_* becomes longer, while that of C*_α_*-N is nearly identical. These results indicate the delocalized radical and cation as observed for the previous system [15,16]. ^1^H-NMR of **2a**^2^⁺ clearly exhibits a downfield shift for the β-proton of pyrrole moiety (δ = 3.53 ppm) compared with that of neutral **2a** (Appendix A), demonstrating global aromaticity of 2a2⁺ as observed for 1a2⁺. The anisotropy of the induced current density (ACID) calculation for 2b2⁺ also supports global aromaticity (Appendix A). Notably, although the chemical shifts of β-protons of 2a2⁺ and 1a2⁺ are almost the same, ethyl protons of **2a**^2+^ appear to be upfield-shifted (CH_2_: Δδ ≈ 0.15 ppm) despite the existence of the electron-withdrawing nitro group. Based on the nucleus-independent chemical shift (NICS) calculation, global aromaticity of **2a**^2+^ seems to be weaker than **1a**^2+^ (Appendix A). Therefore, the slight upfield shifts of ethyl protons of **2a**^2+^ would reflect the weaker global aromaticity, and the similar chemical shift values of β-proton of **2a**^2^⁺ and **1a**^2^⁺ observed would be due to cancellation of the conflicting effects of downfield (NO_2_) and upfield (diminished global aromaticity).
{∑(x*_i_*−<x>)^2^/(n−1)}^1/2^(1)

## 3. Conclusions

The first example of nitration reaction of DEHPHAC **1a** giving nitroDEHPHAC **2a** was presented. The ICT character of **2a** from the electron-rich HPHAC core to the periferal nitro group was confirmed by diagnosis of the solvatochromiclly-shifting absorption centered at ca. 640 nm with TD-DFT calculations. Similar to the DEHPHAC **1a**, nitroDEHPHAC **2a** also showed stable redox properties. Crystallographic analysis of **2a** in not only neutral but radical cationic and dicationic states was performed to reveal a slipped columnar structure for the neutral molecule, π-dimeric structure for the radical cation, and anion-separated structure for the dication. These properties can be applied for supramolecular chemistry and organic electronics such as field effect transistor. Studies including the further derivatization and functionalization of HPHACs are currently underway in our laboratory.

## 4. Materials and Methods

^1^H and ^13^C-NMR spectra were recorded on JEOL (Akishima city, Japan) AL-400 at 400 MHz or Bruker Biospin (Osaka, Japan) AVANCE III at 500 MHz for ^1^H and 100 MHz for ^13^C with use of tetramethylsilane (0 ppm) or residual solvent for ^1^H (7.26 ppm for CDCl_3_) and ^13^C (77.01 ppm for CDCl_3_) signals, and if not otherwise noted, all spectra were measured in CDCl_3_ at 298 K. Cyclic voltammetry (CV) measurements were performed on a CH Instruments-ALS612B electrochemical analyzer using a standard three-electrode cell consisting of Pt working electrodes, a Pt wire counter electrode, and an Ag/AgNO_3_ reference electrode under a nitrogen atmosphere. The potentials were calibrated with ferrocene as an external standard. IR measurements were performed with Thermo Fischer Scientific (Tokyo, Japan) Nicolet iS5 FT-IR. Decomposition points were determined with Büchi M–565 and not corrected. Electronic spectra were recorded on a JASCO (Hachioji, Japan) V–570 UV–VIS/NIR spectrophotometer. High-resolution laser desorption/ionization time-of-flight mass spectra (HR LDI-TOF) were measured on JEOL (Akishima, Japan) JMS-S3000.

All reactions were carried out under nitrogen atmosphere. Thin-layer chromatography (TLC) analyses were carried out using silica gel 60 F_254_ and aluminum oxide 60 F_254_, neutral (Merck Millipore, Tokyo, Japan). Silica-gel chromatography was performed on Silica Gel 60 N (spherical, neutral) purchased from Kanto Chemical Co., Inc. (Tokyo, Japan) Alumina column chromatography was performed on activated alumina (about 200 mesh) purchased from FUJIFILM Wako Pure Chemical Corporation (Osaka, Japan). Dry solvents were purchased from Kanto Chemical Co., Inc. (dichloromethane).

### 4.1. Synthesis of N-[2,3,4,5,6-Pentakis(3, 4-Dimethylpyrrolyl)Phenyl]Pyrrole (DEHPB)

Sodium hydride dispersion in paraffin oil (ca. 60%, 380 mg) was weighed in a round flask, and the oil was removed by dry hexane. After drying, sodium hydride was weighed to be 207 mg (8.63 mmol), and the flask was sealed by a rubber septum. *N*-(2,3,4,5,6-Pentafluorophenyl)-1*H*-pyrrole [15] (180 mg, 0.773 mmol) in dry DMF (10 mL) was added to the flask by a syringe at 0 °C, then 3,4-diethylpyrrole (577 mg, 4.68 mmol) in dry DMF (5.0 mL) was added dropwise at the same temperature, the mixture was stirred, and then, the temperature was raised to 40 °C. After being stirred for 20 h, the reaction mixture was poured into water. The mixture was extracted with ethyl acetate. The organic extract was washed with water, dried over Na_2_SO_4_, and concentrated under a reduced pressure to give a crude product as a white solid. The crude product was purified by trituration with MeOH to give DEHPB as a white powder (468 mg, 0.624 mmol; 81%): ^1^H-NMR (400 MHz) δ 0.93 (m, 30H), 2.21 (q, *J* = 7.4 Hz, 20H), 5.70 (s, 4H), 5.72 (s, 6H), 6.00 (m, 2H), 6.16 (m, 2H); ^13^C-NMR (100 MHz) δ 14.80, 14.82, 14.84, 14.90, 14.93, 14.99, 18.49, 18.51, 110.08, 117.34, 117.45, 121.35, 126.39, 126.57, 126.58, 126.61, 132.13, 133.03, 133.41, 133.86. DI MS *m*/*z* 749 (M^+^+1). IR (ATR) *ν*_max_ = 1494, 1488, 1122, 773, 723 cm^−1^ (Appendix A).

### 4.2. Synthesis of 1,2,3,4,5,6,7,8,9,10-Decaethylhexapyrrolo [2 ,1,5-bc:2′,1′,5′-ef:2′′,1′′,5′′-hi:2′′′,1′′′,5′′′-kl:2′′′′,1′′′′,5′′′′-no:2′′′′′,1′′′′′,5′′′′′-qr][2a,4a,6a,8a,10a,12a]Hexaazacoronene (DEHPHAC 1a)

To a CH_2_Cl_2_ (100 mL) solution of DEHPB (369 mg, 0.492 mmol) was added a CH_3_NO_2_ (5 mL) solution of FeCl_3_ (2.8 g, 18 mmol) at room temperature. After being stirred for 3 h at 60 °C, the mixture was cooled to 0 °C, and then, hydrazine hydrate (5 mL) was added. The mixture was further stirred for 5 min at 0 °C. The mixture was extracted with toluene. The organic phase was washed with water, dried over Na_2_SO_4_, and concentrated under a reduced pressure. The residue was dissolved in a small amount of toluene and then passed through a short column of alumina. The eluate was concentrated to give a reddish brown solid. The solid was recrystallized from toluene/MeOH to give **1a** as orange crystals (250 mg, 0.359 mmol, 67%): ^1^H-NMR (400 MHz, C_6_D_6_) δ 0.85 (m, 30H), 2.09 (q, *J* = 7.4 Hz, 20H), 5.66 (s, 2H); ^13^C-NMR (100 MHz) δ 15.33, 16.98, 17.18, 17.87, 17.99, 18.05, 104.95 (2 kinds of C), 105.63, 106.25, 106.30, 106.60, 117.61, 117.98, 118.48, 118.77, 120.20, 122.01, 122.56, 122.72, 122.75, 122.99. HR-LDI-TOF MS: calcd. for C_50_H_52_N_6_^+^: 736.4253; found: 736.4287. IR (ATR) *ν*_max_ = 1630, 1448, 1371, 1360, 744 cm^−1^ (Appendix A).

### 4.3. Synthesis of nitroDEHPHAC **2a**

Dichloromethane (50 mL) was added to a mixture of DEHPHAC **1a** (49 mg, 0.067 mmol) and AgNO_2_ (100 mg, 0.65 mmol). The suspension was stirred vigorously at 20 °C for 1.5 h. After addition of hydrazine hydrate (ca. 1 mL), the solution was concentrated under a reduced pressure. A minimum solution of the residual solid in toluene was put on a silica-gel column filled with hexane, and then, the column was eluted with toluene. Green fractions were combined and then concentrated in vacuo to give **2****a** (27 mg, 0.034 mmol, 51%) as a dark green solid: ^1^H-NMR (400 MHz) δ 6.47 (s, 1H), 2.37–2.78 (m, 20H), 1.11–1.27 (q, 27H), 1.05 (t, 3H, ^3^*J* = 7.3 Hz). ^13^C-NMR (125 MHz) δ 133.75, 131.91, 126.19, 125.62, 125.00, 124.43, 124.10, 123.92, 123.79, 123.67, 122.87, 121.61, 120.73, 119.77, 119.35, 118.79, 108.25, 107.84, 106.60, 106.44, 106.29, 103.60, 102.02, 19.171, 18.14, 17.99, 17.657, 16.980, 16.87, 16.47, 15.31, 15.02. (The signals due to core benzene carbons were not found). HR-LDI-TOF MS: calcd. for C_50_H_51_N_7_O_2_^+^: 782.4104; found: 782.4106. IR (ATR) *ν*_max_ = 1626, 1360, 1309, 1275, 1240 cm^−1^ (Appendix A).

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
