# Peer review of "Synthesis and Properties of NitroHPHAC: The First Example of Substitution Reaction on HPHAC"

_molecules, 2020, doi:10.3390/molecules25112486_

Round 1

Reviewer 1 Report

The manuscript from Takase, Uno and co-workers, describes the first example of electrophilic substitution on a HPHAC core. The scope of the manuscript is limited to just one reaction (and just one new molecule) but the work is detailed and nicely presented. The crystal structures are particularly interesting, notably the pi dimerization observed for the radical cation (Shinokubo’s recent example, 10.1021/jacs.8b00798, should be cited here).

A potential shortcoming of the manuscript is the lack of synthetic/analytical details for 1a/1b. These are new compounds and the authors promise to provide details in a forthcoming publication. While I trust the authors, the usual practice is to characterize all new compounds or at least provide the missing data in a review-only material. It is up to the editors to decide if these data are required for publication. For me, the manuscript is publishable subject to minor revisions noted herein.

Other comments

In the introduction, references on edge-expanded and chiral HPHACs are missing (10.1021/ja508963v, 10.1039/C5SC03280F, 10.1002/anie.201900175, 10.1021/jacs.9b13942).

“Basis set: B3LYP/6-31G(d,p)”
this is more than a basis set

“asymmetric cell,”
the asymmetric unit or the unit cell, please revise

“Molecules in the dimer face by the convex side.”
Style

“Notably, although the chemical shifts of β-protons of 2a2⁺ and 1a2⁺ are almost the same, those of ethyl protons of 2a2+ appears to be upfield-shifted (CH2: Δδ ≈ 0.15 ppm) despite the existence of the electron-withdrawing nitro group (Figure S5).”
This could be explained by noting that for the beta-pyrrolic H, the downfield shift caused by the NO2 group is mostly cancelled by the upfield relocation caused by the diminished global aromaticity

“On the other hand, intramolecular hydrogen-bonding between the nitro group and the β-proton of pyrrole moiety induces the low-field shift in the 1H NMR spectrum”
This is caused by magnetic anisotropy of the NO2 group and not by hydrogen bonding.

Author Response

 According to your advise, we added some explanations with references in the introduction. Therefore, the reference numbering has been updated.

 In order to clarify the compounds we prepared, Scheme 1 has been changed. The structures of compounds 1b and 2b are just utilized for calculation and not prepared. According to this change, some explanations were changed. Experimental details for the preparations of hexapyrrolylbenzene, DEHPHAC 1a, and nitroDEHPHAC 2a were mentioned in the experimental section.

Thank you for your indication of typographical errors. We corrected the errors.

Our plausible reaction mechanism giving nitroDEHPHAC is stated briefly in the text, although the mechanism is not examined and unclear at this moment.

Reviewer 2 Report

Readers of the communication might be curious how the authors arrived at 9.7 molar excess ratio of AgNO2 for HPHAC nitration, to achieve 51% yield of the product. Were there any attempts for yield optimization? Experimental states that the substrates were dissolved in CH2Cl2 at room temperature. Is AgNO2 really that soluble - 100 mg in 50 mL?

Author Response

We haven't optimized reaction conditions of the nitration giving 2a. The reaction was carried out in a suspension by mixing 1a in dichloromethane with undissolved silver nitrite. 

Reviewer 3 Report

Review of Manuscript ID molecules-779434

The manuscript by Takase, Uno, and coworkers reports the synthesis of a hexapyrrolohexaazacoronene (HPHAC) derivative containing one nitro substituent. HPHAC is an aromatic compound that has a benzene ring surrounded by 6 pyrroles fused to one another.

The product obtained was characterized by 1H and 13C NMR, X-ray crystallography, high resolution TOF MS and UV-vis spectroscopy. Computational studies of the compound were performed using B3LYP/6-31G(d,p) to predict various molecular properties including geometry, molecular dipole and UV-vis absorption via TD-DFT method. NitroHPHAC 2a adopts slipped columnar structure in the solid state based on the X-ray crystallographic data with cofacial pi-stacking interplanar distance of ~3.4 angstrom.

Cyclic voltammetry (CV) and differential pulse voltammograms (DPV) clearly demonstrate two sequential oxidation processes are accessible for compound 2a. Chemically mediated oxidation via SbCl5 and UV-vis studies reveals isosbestic points upon single and double oxidation processes. X-ray crystallographic data of the oxidized salt (2a•+ and 2a+2) was also obtained illustrating a decrease in the interplanar pi-stacking distance to ~3.34 angstrom. NMR spectral data also supports a change in the electronic properties and change in induced current density as a result of oxidation, namely a downfield shift in the beta-hydrogen on the pyrrole moiety.

The manuscript is well written and provides clear details of the experimental work and results obtained. The structures and the X-ray crystallographic data are very interesting results and will be of high interest to readers of Molecules and researchers in the field of organic electronics.

The manuscript would benefit from several minor changes outlined below:

Line 93: Figure 4 caption should specify the supporting electrolyte (and its concentration) used to measure the cyclic voltammetry data. The Supporting Information should provide additional details regarding those experiments including the type of working electrode (glassy carbon or Pt or other material) and reference electrode used as well as what counter electrode was used (Pt coil?).

Line 163-164: “The residue was purified by silica-gel column chromatography using hexane and toluene as an eluent to give the dark green solid of 2a…” Can the authors provide a ratio of the hexane to toluene used? Is the column run isocratic or as a gradient from 100% hexanes to 50%:50% hexanes:toluene? This detail will be important for future scientists who may want to use their procedure to synthesize/purify compound 2a.

Supporting Information page S15: The authors provide an ESP surface for neutral structure 2a. Could the authors provide the analogous ESP surface for mono and di-oxidized structures 2b•+ and 2b+2. As well as include them in the bond length comparison shown in Figure 8b in the Supporting Information to demonstrate the effect of oxidation on the bond length. From the Supporting Information (page S17 and S18), these structures have already been calculated so it should be minimal additional work to address these two comments.

The final comment to the authors is that they should either provide a .cif file containing the three crystallographic structures as Supporting Information and/or deposit the X-ray crystallographic data in The Cambridge Crystallographic Data Centre (CCDC) database (https://www.ccdc.cam.ac.uk/support-and-resources/ ) so the reader can have access to the three dimensional structures. I saw that CCDC numbers were listed in the Supporting Information but I did not have access to those files (X-ray crystallographic data).

Author Response

 According to your suggestion, we added the conditions of electrolysis in the text and experimental part.

 Information for the column chromatography of 2a is added in the experimental part.

 In SI, illustration of ESP surfaces of 1b is added. Merging of the ESP drawing and bond lengths from X-ray analysis made the figure very unclear. Therefore, we listed them separately in SI.

  If you want to check the X-ray data we have deposited to CCDC, you can request the data to CCDC by e-mail.